# OpenReview forum: "TokenSwap: Backdoor Attack on the Compositional Understanding of Large Vision-Language Models"
_ICML.cc/2026/Conference — ICML 2026 regular_

### Official Review · Reviewer_AKHf · 2026-03-05

**Soundness:** 2
**Presentation:** 3
**Significance:** 3
**Originality:** 3
**Overall Recommendation:** 4
**Confidence:** 4

**Summary:**

This paper proposes TokenSwap, a backdoor attack on LVLMs that targets compositional understanding rather than forcing a fixed target phrase. The key is to implant a trigger-conditioned “bag-of-words”-like failure mode: when a visual trigger is present, the model still mentions the right entities in the image but misstates their relationships, operationalized primarily as swapping grammatical roles in the generated answer.

**Compliance With Llm Reviewing Policy:**

Affirmed.

**Final Justification:**

They have addressed my original concerns very well, and they also conducted additional experiments on the newly raised questions to further resolve my concerns.

**Key Questions For Authors:**

1.How sensitive is the reported ASR to the choice of LLM judge and prompt?

2.How does TokenSwap perform under more realistic trigger constraints (small, low-contrast, physically plausible, varying locations), and does ATW still provide the same gain?

3.Does the attack perform under smaller poisoning budgets (e.g., 1% or below)? The rate mentioned in paper is still high in real world.

4.Since TokenSwap specifically targets compositional relations, it would be helpful to analyze whether the backdoored model introduces false relational swaps on clean inputs. Metrics such as BLEU and ROUGE may not capture such errors. For example, “A dog is chasing a man” and “A man is chasing a dog” would still yield high lexical overlap despite reversing the semantic relation.

**Limitations:**

yes

**Strengths And Weaknesses:**

Strengths

1.The threat model and objective are clearly articulated, and the attack mechanism is concretely defined via subject–object swapping and poisoned dataset construction.

2.The ATW loss is a reasonable, targeted intervention for the stated learning challenge, and ablations on its hyperparameters are provided.

3.The paper highlights a practically relevant and arguably underexplored security risk: backdoors that preserve lexical plausibility and avoid fixed malicious strings, making them harder to catch with simple perplexity- or rule-based detectors and potentially more dangerous in safety-critical deployments.

Weaknesses

1.ASR measurement relies on an LLM-as-judge pipeline (the paper uses GPT-4o-mini to detect whether a swap occurred). Even with reported agreement numbers, this introduces evaluator dependence and potential failure modes (edge cases, paraphrases, multi-token role changes) that are hard to fully characterize without releasing evaluation prompts, calibration sets, and error analysis.

2.The attack is mainly instantiated as subject–object swapping, but the paper also claims broader emergent distortions. Those broader behaviors are compelling, yet they would benefit from more systematic quantification beyond illustrative examples.

---

> ### Author Rebuttal · Authors · 2026-03-31
>
> Thank you for recognizing the novelty of our threat model and proposed method.
>
>
> **W1 (LLM-as-judge pipeline):** Thanks for the concerns.
>
> (1) ***Evaluator dependence.*** We re-evaluated the same triggered outputs on Flickr8k using different LLM judges. The results are highly consistent (**std=0.66%**), indicating that the evaluation is not sensitive to the specific judge used, which may stem from the fact that the judge is provided with full oracle information, including the ground-truth caption, the image, and the definition of TokenSwap behavior (Figure 8).
>
> |Evaluator|ASR|
> |-|-|
> |GPT-4o-mini|87.50%|
> |GPT-4o|87.25%|
> |Claude-Sonnet 4.6|88.75%|
> |Mean +/- Std|87.83% +/- 0.66%|
>
>
> (2) ***Potential failure modes.*** We further examined potential failure modes through human validation. On MSCOCO, GPT-4o-mini achieves a **97.3% agreement rate** with independent human annotators, suggesting that such failure cases are limited in practice. The remaining 2.7% disagreement cases fall into three identifiable categories:
> - Implicit subjects: both entities are implied, making the swap ambiguous even to humans.
> - Near-synonymous roles: entities are interchangeable (e.g., "two dogs playing"), so the swap is semantically equivalent.
> - Truncated outputs: incomplete captions lack sufficient context for swap determination.
>
> Moreover, **the false positive rate of the GPT-4o-mini judge is 0%** across more than 15 configurations. Full evaluation prompts are provided in Figure 8. We will add calibration sets and detailed error analysis in the revised paper for reproducibility.
>
> **W2 (Systematic quantification of emergent distortions):** Thanks for your wonderful suggestions! Although the model is trained only with single-token subject–object swaps, it induces richer emergent behaviors at test time, which we systematically quantify as follows (one output may exhibit multiple behaviors):
> |Emergent Behavior|ratio|Example|
> |-|-|-|
> |***Beyond single token swap***|||
> |Multi-token phrase swap|64.5%|"a snowboarder doing a trick" → "a trick doing a snowboarder"|
> |***Beyond lexical swap***|||
> |Voice/Preposition shift|30.5%|"A chases B" → "B is chased by A"|
> |***Unintended compositional swap***|||
> |Attribute migration: properties attach to the wrong entity|7.8%|"a boy in a yellow shirt holding sparklers" → "a sparkler in a yellow shirt holding a boy"|
> |Role fabrication: novel relational structure not in ground-truth|44.1%|"a man riding a horse" → "a horse carrying a rider while being guided by a man"|
>
>
> **Q1 (Prompt sensitivity):** We evaluated the same triggered outputs on Flickr8k using four prompt variants with GPT-4o-mini. As shown below, **ASR std = 0.39%**. Combined with the cross-evaluator results in W1 (**std = 0.66%** across three LLM judges), this indicates that the reported ASR is robust to both prompt wording and evaluator choice.
>
> |Prompt Variant|ASR|
> |-|-|
> |Original (Figure 8)|87.50%|
> |Paraphrase 1 (simplified)|87.50%|
> |Paraphrase 2 (task-agnostic)|86.59%|
> |Paraphrase 3 (more specific)|87.50%|
> |**Mean +/- Std**|87.27% +/- 0.39%|
>
>
> **Q2 (Realistic trigger constraints):**  Thanks for the question. We further evaluate TokenSwap under more realistic trigger constraints. As shown below, TokenSwap remains effective across all settings and consistently outperforms BadNet, confirming ATW's benefit generalizes to realistic trigger constraints.
>
> |Trigger Config|TokenSwap ASR|BadNet|
> |-|-|-|
> |BadNet of size 10px (small)|88.28%|54.69%|
> |Type WaNet warping (low-contrast)|92.19%|59.38%|
> |Left Top (varying locations)|92.19%|64.06%|
> |Refool (physically plausible)|84.06%|56.25%|
>
> [1] Reflection Backdoor: A Natural Backdoor Attack on Deep Neural Networks, ECCV 2020
>
>
> **Q3 (Smaller poisoning budgets):** Thanks for the concern. We further evaluate TokenSwap on Flickr8k with BLIP2 under smaller poisoning budgets. As shown below, the ASR degrades gracefully as the poisoning rate decreases: even at 1%, TokenSwap still achieves 75.20% ASR, which remains substantially higher than the baselines (e.g., 9.38% / 21.88% ASR for TrojVLM / VLOOD under the same setup). **TokenSwap remains practical even under realistic low-budget poisoning scenarios**.
>
> |Poisoning Rate|ASR|
> |-|-|
> |50%|91.41%|
> |10%|88.33%|
> |5%|86.68%|
> |3%|84.12%|
> |1%|75.20%|
>
>
> **Q4 (False relational swaps on clean inputs):** Thanks for raising this concern. We agree that BLEU/ROUGE cannot capture compositional swap errors. To directly assess this concern, we provide two pieces of evidence: (1) **The clean-input ASR is 0% across all experiments**, indicating that the backdoor is activated only with trigger; and (2) We report the SPICE Relation F1, which evaluates relational tuple (subject–verb–object) correctness. The backdoored model scores **0.1144**, comparable to the clean baseline **(0.1218)**, suggesting no meaningful increase in false relational swaps on clean inputs.

---

> > ### Author Rebuttal · Reviewer_AKHf · 2026-04-02
> >
> > Thank you for your responses and the additional experiments. They have all solved my concerns. I am willing to increase my score to 4.

---

> > > ### Author Response · Authors · 2026-04-02
> > >
> > > Thank you for the important follow-up question. We agree that investigating the robustness of TokenSwap under downstream adaptation is crucial, as backdoored models in practice are likely to be further fine-tuned before deployment.
> > >
> > > To address this, we conduct additional experiments using BLIP2 as the victim model, with MSCOCO as the poisoning dataset and Flickr8k/Flickr30k as out-of-domain (OOD) data. For the fine-tuning experiments, we fine-tune the backdoored BLIP2 using 2,000 clean image-caption pairs for 16 epochs. We evaluate four settings: in-domain evaluation (ID-Zeroshot), out-of-domain evaluation (OOD-Zeroshot), in-domain fine-tuning (ID-Finetuning), and out-of-domain fine-tuning (OOD-Finetuning). Results are shown below:
> > >
> > > | Setting | ID-Zeroshot | OOD-Zeroshot | ID-Finetuning | OOD-Finetuning |
> > > |---|---|---|---|---|
> > > | MSCOCO→Flickr8k (BadNet) | 81.25% | 54.69% | 51.72% | 33.83% |
> > > | MSCOCO→Flickr8k (TokenSwap) | **91.41%** | **88.28%** | **62.39%** | **59.63%** |
> > > | MSCOCO→Flickr30k (BadNet) | 53.13% | 44.53% | 51.72% | 23.15% |
> > > | MSCOCO→Flickr30k (TokenSwap) | **80.47%** | **76.56%** | **62.39%** | **54.09%** |
> > >
> > > (Note that ID-Finetuning results are identical across OOD targets, as in-domain fine-tuning and evaluation are both conducted on MSCOCO, independent of the OOD dataset.)
> > >
> > > We observe that downstream fine-tuning reduces ASR for both attacks, indicating that adaptation can partially weaken backdoor behavior. However, TokenSwap consistently remains substantially more effective than BadNet across all four settings. In particular, after OOD fine-tuning, TokenSwap still retains 59.63%/54.09% ASR on Flickr8k/Flickr30k, compared with 33.83%/23.15% for BadNet.
> > >
> > > Overall, these results suggest that the backdoor induced by TokenSwap is more persistent under downstream adaptation than BadNet. They also indicate that TokenSwap is more resistant to removal under both in-domain and out-of-domain clean fine-tuning.
> > >
> > > We hypothesize that this robustness may arise from two factors. First, the ATW loss explicitly reinforces swapped compositional relations, which may lead to a stronger trigger-behavior association and make the attack harder to forget during clean fine-tuning. Second, TokenSwap is instance-dependent rather than tied to a fixed target output, which may spread the backdoor effect more broadly across the model and thus improve its persistence under downstream adaptation.

---

### Official Review · Reviewer_tsXZ · 2026-03-08

**Soundness:** 4
**Presentation:** 4
**Significance:** 3
**Originality:** 4
**Overall Recommendation:** 4
**Confidence:** 4

**Summary:**

This work presents TokenSwap, a backdoor attack against LVLMs that specifically targets object relationship understanding. By swapping subjects and objects in prompts, TokenSwap compromises visual-language alignment while ensuring semantic coherence. Furthermore, an Adaptive Token-Weighted (ATW) Loss is introduced to dynamically weight swapped tokens according to prediction confidence.

Extensive experiments were conducted on multiple LVLMs (BLIP-2, InstructBLIP, LLaVA-1.5, Qwen-VL) and datasets (Flickr8k/30k, MSCOCO, VQAv2, OKVQA), including cross-dataset and cross-task evaluations.

**Compliance With Llm Reviewing Policy:**

Affirmed.

**Final Justification:**

The authors have successfully addressed all concerns regarding the ATW loss behavior, human inspection details, and the specific scope of attack evasiveness through comprehensive additional data and refined claims. Given the technical soundness and the novelty of the identified vulnerability in LVLM compositional understanding, I maintain the initial score.

**Key Questions For Authors:**

Please refer to the weaknesses.

**Limitations:**

yes

**Strengths And Weaknesses:**

Strengths：
1. The paper is well-written and easy to follow.

2. The paper identifies a novel backdoor vulnerability targeting the compositional understanding of LVLMs, which is significant for the community.

3. The ATW loss is an effective and ingenious design for instance-dependent backdoor attacks.

4. The paper conducts comprehensive experiments, including cross-dataset and cross-task evaluations, and benchmarks against various existing defense methods.

Weaknesses：
1. The paper lacks an analysis of the balancing strategy for the ATW loss function under varying poisoning rates. It remains unclear whether a high poisoning rate would adversely affect the generation of unswapped tokens.

2. The paper provides insufficient details regarding human inspection. Including specific failure cases where human observers successfully detected the attack would strengthen the evaluation.

3. The evasiveness of the method may be overstated. Certain counterintuitive outputs should be detectable by LLMs or VLMs even without reference to the image, such as "Two kangaroos are petting a woman." in Figure 7.

---

> ### Author Rebuttal · Authors · 2026-03-31
>
> Thank you for the positive assessment, recognizing the novel vulnerability, ingenious ATW design, and comprehensive experiments.
>
>
> **W1 (ATW behavior under varying poisoning rates):** Thanks for raising this concern. As shown below, increasing the poisoning rate does not materially degrade clean-sample generation. Even at **90%** poisoning, **R-1 / BLEU** decrease by only **0.83 / 0.56** relative to the clean model, and this degradation remains small across all poisoning rates rather than increasing systematically. Meanwhile, **clean-input ASR remains 0%** in all settings, suggesting that TokenSwap preserves clean-input behavior well even under high poisoning rates.
>
> | Poisoning Rate | ASR (Poisoned) | R-1 (Clean) | BLEU (Clean) | ASR (Clean) |
> |---|---:|---:|---:|---:|
> | Clean model (no attack) | — | 45.69 | 13.64 | 0% |
> | 10% | 75.00% | 45.23 | 12.86 | 0% |
> | 30% | 83.59% | 45.37 | 13.75 | 0% |
> | 50% (default) | 91.41% | 45.51 | 12.79 | 0% |
> | 70% | 92.97% | 44.82 | 13.08 | 0% |
> | 90% | 92.97% | 44.86 | 13.08 | 0% |
>
>
> **W2 (Insufficient details regarding human inspection):** Thank you for this valuable suggestion. Our manual inspection shows that successful human detection is concentrated in cases where the swapped entities produce clearly unnatural descriptions (e.g., "kangaroos petting a woman"). In contrast, when the swapped entities are semantically compatible and their roles are plausibly interchangeable (e.g., "a woman walking behind a man" → "a man walking behind a woman"), the attack becomes much harder for human observers to detect. This suggests that TokenSwap’s stealth increases with the semantic plausibility of the swapped roles. We will include both representative success cases and failure cases from human inspection in the revised paper.
>
>
>
>
> **W3 (Evasiveness of TokenSwap may be overstated):** Thanks for the concern. We agree that clearly counterintuitive outputs (e.g., "two kangaroos are petting a woman") can often be detected even without image reference. Our evasiveness claim specifically targets the **plausible-swap setting**: swaps between semantically close, role-interchangeable entities where neither direction violates commonsense, making the output difficult to distinguish from a normal caption without consulting the image. Critically, such cases arise naturally in safety-critical settings. For example, "a pedestrian yielding to a vehicle" versus "a vehicle yielding to a pedestrian" is grammatically valid in both directions, yet implies opposite driving decisions. We will revise the paper to (1) explicitly restrict the evasiveness claim to semantically close, role-interchangeable swaps, (2) acknowledge clearly counterintuitive swaps (e.g., animal–human role reversal) as a limitation, and (3) emphasize the safety-critical implications of the plausible-swap setting as the primary threat model.

---

> > ### Author Rebuttal · Reviewer_tsXZ · 2026-04-03
> >
> > Thanks for the authors' effort on the response. I'll keep my score.

---

> > > ### Author Response · Authors · 2026-04-03
> > >
> > > We are glad that our rebuttal has fully resolved your concern. Thank you again for acknowledging and being positive about our work!

---

### Official Review · Reviewer_YZ4X · 2026-03-08

**Soundness:** 3
**Presentation:** 2
**Significance:** 2
**Originality:** 3
**Overall Recommendation:** 3
**Confidence:** 4

**Summary:**

This paper proposes TokenSwap, a backdoor attack on large vision-language models that aims to corrupt compositional understanding rather than forcing a fixed target string. The attack poisons a subset of training samples by stamping a visual trigger on images while swapping subject and direct-object roles in the corresponding textual outputs, encouraging bag-of-words style generation under trigger. Experiments across several LVLMs and datasets report higher attack success rates and improved evasiveness relative to fixed-pattern backdoors.

**Compliance With Llm Reviewing Policy:**

Affirmed.

**Key Questions For Authors:**

1.	The paper mainly demonstrates subject–object swaps in relatively simple captions. Does the attack still succeed when captions involve more complex syntactic structures, such as passive voice, multiple objects, or nested clauses?
2.	How effective is TokenSwap in other multimodal tasks that require deeper reasoning, such as visual grounding, instruction following, or multimodal dialogue?
3.	In Table 1, TokenSwap sometimes outperforms the clean model in model utility. Could you explain it?
4.	How sensitive is the attack to the location and size of the visual trigger?
5.	Are there any insights into practical defense mechanisms against TokenSwap?

**Limitations:**

yes

**Strengths And Weaknesses:**

Strengths
1.	The paper is well-written and easy to follow.
2.	Clear and focused attack objective: instead of fixed phrases, the backdoor target is a semantic failure mode, which is a realistic vulnerability class for LVLM deployments.
3.	The method is simple and reasonable. In particular, the ATW loss is easy to implement and intuitively connected to the “hard-to-learn swapped tokens” issue

Weaknesses
1.	Limited evaluation across model architectures. Since different architectures may exhibit different sensitivities to token-level poisoning, it is unclear whether TokenSwap generalizes consistently across a broader range of multimodal models, especially more recent vision–language models like Qwen-3-VL. A more diverse evaluation would strengthen the empirical claims.
2.	Limited interpretability analysis of the attack mechanism. While the paper demonstrates the effectiveness of TokenSwap empirically, it provides limited analysis of why the attack works from a model representation perspective. For example, it would be useful to study how the poisoning samples alter the alignment between visual tokens and language tokens, or how the model’s internal attention patterns change when the trigger is present.

---

> ### Author Rebuttal · Authors · 2026-03-31
>
> Thank you for recognizing the novelty of our attack objective and proposed method.
>
> **W1 (TokenSwap with new model architectures):** Thanks for the suggestions. We provide the full results across all tested architectures below, including the recent **Qwen3-VL** as well as **InternVL2.5**. The results show that TokenSwap remains effective across diverse LVLM architectures.
> |Model|Connector|LLM Backbone|ASR|
> |-|-|-|-:|
> |BLIP-2 |Q-Former|OPT|91.41%|
> |InstructBLIP|Q-Former|Vicuna|89.06%|
> |LLaVA-1.5 |MLP|Vicuna|82.81%|
> |InternVL2.5|MLP|InternLM2.5|80.94%|
> |Qwen3-VL|MLP|Qwen3|84.38%|
>
> **W2 (Why TokenSwap works via a representation perspective):** Thanks for your insightful suggestion. We conduct two analyses to understand TokenSwap's mechanism at the representation level.
>
> (1) ***Vision-text alignment preservation.*** We compare TokenSwap with **TokenReplace** that replaces tokens rather than swapping them:
> |Condition|CLIP Similarity|Difference from Clean|
> |-|-:|-:|
> |Clean|0.2716|-|
> |TokenSwap|0.2718|**0.074%**|
> |TokenReplace|0.2386|**12.15%**|
>
> TokenSwap preserves vision-text alignment almost perfectly, whereas TokenReplace substantially degrades it. This supports our core insight: because LVLM visual encoders often **weakly encode compositional relationships**, TokenSwap can succeed without significantly disrupting CLIP-level semantic alignment.
>
> (2) ***Attention and logit analysis around swapped-token generation.*** We further examine whether TokenSwap operates through attention redirection, as is common in prior backdoor attacks, by analyzing attention maps immediately before the swapped tokens are generated:
> |Metric|Value|
> |-|-:|
> |Trigger attention/average attention|0.90|
> |Logit drop when trigger patch is masked|−0.031±0.194|
> |Logit drop when random patch is masked|−0.031±0.133|
>
> The trigger receives even less attention than average, and masking it causes no larger logit change than masking a random patch. This suggests that the trigger effect is absorbed into internal representations early, rather than being continuously mediated through attention. This indicates that **TokenSwap does not rely on attention hijacking. Instead, it exploits the weakness in the model’s compositional representation**. This distinguishes TokenSwap from typical attention-hijacking triggers and further highlights compositional insensitivity as a particularly exploitable vulnerability surface.
>
> **Q1 (TokenSwap with complex syntax):** We evaluated TokenSwap on syntax-specific subsets of Flickr8k. The results show that TokenSwap remains effective even when captions contain more complex syntactic structures.
> |Syntax Type|# Samples|ASR|
> |-|-:|-:|
> |Passive voice|34|**100.0%**|
> |Multiple objects (>1 dobj/pobj)|139|**91.4%**|
> |Nested clauses (relcl/advcl/ccomp/xcomp)|99|**90.9%**|
> |Simple syntax|105|**92.4%**|
>
> **Q2 (Other multimodal tasks):** We evaluated TokenSwap on tasks requiring deeper reasoning (the first three use Flickr8k images and details will be updated in revised paper):
> |Task|ASR|
> |-|-:|
> |**Visual grounding (relational description)**|**60.82%**|
> |Instruction following (constrained QA)|**60.61%**|
> |Multi-turn dialogue|**58.97%**|
> |VQAv2|**84.83%**|
> |OKVQA|**78.92%**|
>
> TokenSwap consistently induces compositional distortion across all tasks, with ASR ranging from 58.97% to 84.83%. The relatively lower ASR on the first three tasks is due to the output format constraints (e.g., single-entity or yes/no answers) that limit token positions where swaps can manifest.
>
> **Q3 (why TokenSwap utility > clean model sometimes):** Thanks for the question. Backdoor attacks are designed to establish a spurious trigger→swap shortcut, rather than systematically altering the model’s behavior on clean inputs. Since the backdoored and clean models are both trained on the same clean data distribution (as described in Sec. 2.3), the slight utility improvement over the clean model is best understood as normal fine-tuning variance rather than a statistically meaningful gain caused by the attack.
>
> **Q4 (Trigger sensitivity):** TokenSwap is robust with both trigger size and location (Table 10):
> |Dimension|Variant|ASR|
> |-|-|-|
> |**Size**|10px|88.28%|
> ||30px|90.63%|
> ||50px|90.63%|
> |**Location**|Top-left|87.50%|
> ||Four Corners|92.19%|
> ||Middle|89.06%|
>
>
> **Q5 (Adaptive defense insight):** Thanks for this point. A principled adaptive defense is **post-hoc swap-back detection**: (1) parse the output caption and enumerate candidate inverse swaps; (2) use a clean VLM (e.g., GPT-4V) as a judge to score which version better matches the image. If the swapped-back version receives a higher score, the output can be flagged as potentially backdoored. However, this defense requires **K** additional multimodal inference calls per output (one for each candidate swap), which introduces substantial overhead in both latency and cost. The core challenge is that TokenSwap is distribution-preserving: the same tokens still appear in the output, but in swapped compositional roles.

---

> > ### Author Rebuttal · Reviewer_YZ4X · 2026-04-07
> >
> > Thanks for author's effort for the rebuttal.

---

> > > ### Author Response · Authors · 2026-04-07
> > >
> > > Thank you for your feedback on our rebuttal. We noticed that you selected "Partially resolved or unresolved" in your acknowledgement. Could you please let us know which specific concerns remain unaddressed? We are committed to resolving them promptly and would appreciate any further guidance to improve our work.

---

### Official Review · Reviewer_WrLv · 2026-03-13

**Soundness:** 3
**Presentation:** 3
**Significance:** 2
**Originality:** 2
**Overall Recommendation:** 4
**Confidence:** 4

**Summary:**

The paper proposes TokenSwap, a backdoor method on VLM that injects a visual trigger into samples during training while swapping the subject and object of the caption to shift the semantic meaning of the sentence with incorrect compositional relationship. By using an adaptive token-weighted loss, the target model learns to associate the trigger with these swapped tokens.

**Compliance With Llm Reviewing Policy:**

Affirmed.

**Final Justification:**

Most of my concerns are resolved.

**Key Questions For Authors:**

1. Could the authors also show the cross-modal evaluation of the LLaVA-7B model, since BadNet and TokenSwap are closer in the in-modal setting with LLaVA-7B as the base model?

2. The adaptive defense is relatively weak. Could the authors also showcase some backdoor detection methods against TokenSwap?

3. The datasets also contain simple captions. Can the authors test the method on datasets with longer and more complex captions (such as ShareGPT4V)?

**Limitations:**

Limitation not addressed, but mentioned social impact of the work. However, weakness 3 is not fully resolved and the method's novelty is limited, so I choose to give the score of borderline accept.

**Strengths And Weaknesses:**

Strengths:

1. The paper is technically sound.

2. The follow of the paper is clear and easy to follow.

3. The experiment section are done thoroughly

Weaknesses:

1. I don't think the attack itself is significant enough. While switching the subject and object can make the sentence confusing, they are easy to correct and do not influence the meaning of the sentence significantly.

2. The paper is also not very novel. The data poisoning idea is not fundamentally new. The only method is the adaptive token weighted loss, which is not a significant change from the standard BadNet poisoning.

3. BadNet performs quite well in Tables 1 and 2: the ASR gap between BadNet and TokenSwap is small for LLaVA-7B, and the utility scores are also similar for both attacks. Additionally, BadNet obtains a higher utility score in Table 2 compared to TokenSwap.

4. Could the authors also show the cross-modal evaluation of the LLaVA-7B model, since BadNet and TokenSwap are closer in the in-modal setting with LLaVA-7B as the base model?

5. The adaptive defense is relatively weak. Could the authors also showcase some backdoor detection methods against TokenSwap?

6. The datasets also contain simple captions. Can the authors test the method on datasets with longer and more complex captions (such as ShareGPT4V)?

---

> ### Author Rebuttal · Authors · 2026-03-31
>
> Thank you for acknowledging our technical soundness, clear writing, and thorough experiments!
>
> **W1 (Significance of TokenSwap)**: Thanks for your concern. We agree that TokenSwap preserves grammatical fluency and often retains the same lexical items, which may make TokenSwap appear mild. However, this is precisely what makes it significant. Consider a safety-critical scenario: in medical image reporting, "tumor compressing nerve" versus "nerve compressing tumor" reverses the clinical finding while sharing all the same words. What makes such distortions even more dangerous is that they are exceptionally hard to catch.
>
> (1) ***Serious meaning distortion***. Although triggered and clean outputs retain noticeable lexical overlap, pairwise semantic evaluation by GPT-4o-mini on triggered/clean output pairs shows that **93.0% are semantically non-equivalent**: 39.7% are explicit role reversals and 53.3% exhibit other meaning changes.
>
> (2) ***Hard to detect/correct***. Because triggered outputs remain grammatically correct and mention the same entities, automated detection largely fails. Under realistic conditions, **84.5% of TokenSwap’s malicious outputs pass undetected, whereas almost all outputs from conventional attacks are detected** (Table 5). Moreover, because TokenSwap produces input-dependent distortions rather than a fixed pattern, perplexity-based detection also fails (Figure 1). TokenSwap is thus difficult to correct since it is difficult to detect.
>
> **W2 (Novelty):** Thanks for this concern. We agree that the overall poisoning pipeline follows the standard backdoor paradigm. However, our novelty is not a new generic poisoning recipe (***how to attack***), but a new attack surface in LVLMs (***what to attack***): weak visual grounding of compositional relations. Empirically, compositionally different inputs can remain extremely close in representation space (cosine similarity 0.902–0.995, Table 12), whereas ordinary semantic changes produce much larger shifts (0.317, Table 12). TokenSwap is tailored to this vulnerability: instead of enforcing a fixed malicious target, it corrupts relational semantics while preserving lexical plausibility. Moreover, to further remind the model of the subtle change in the poisoned samples and override its language prior, we propose ATW loss to specifically upweights the swapped relational tokens.
>
> In summary, our contribution is twofold: (1) identifying compositional insensitivity in LVLM visual representations as a new backdoor attack surface, and (2) proposing a compositional backdoor objective, instantiated with ATW loss, that specifically exploits this vulnerability.
>
> **W3 (Small gap for LLaVA):** Thanks for your observation. (1) We agree that the margin over BadNet is smaller on LLaVA-7B (**5%–7%**), but this remains a meaningful gain, and TokenSwap achieves substantially larger improvements on other architectures: **BLIP2 (+27%)**, **InstructBLIP (+35%)**, and **Qwen2.5-VL (+28%)**. We therefore view the LLaVA-7B result as a small-margin case, rather than evidence against TokenSwap’s effectiveness. (2) Regarding utility, our goal is not to outperform BadNet on every clean metric, but to preserve clean utility while achieving stronger attack effectiveness. Across five models, TokenSwap's clean utility is nearly identical to the clean model (**R-1: +0.01, R-L: +0.06, BLEU: +0.47**), confirming it achieves stronger attack effectiveness without degrading clean performance.
>
> **W4 (Cross-dataset performance, LLaVA):** Thanks for your question. We additionally report the cross-dataset results for LLaVA-7B (Table 8). TokenSwap shows a clear advantage over BadNet in the cross-dataset setting, with **+11.72%** ASR on **MSCOCO → Flickr8k** and **+10.94%** on **MSCOCO → Flickr30k**.
> |Setting|TokenSwap ASR|BadNet ASR|Gap|
> |-|-:|-:|-:|
> |MSCOCO→Flickr8k|80.47%|68.75%|**+11.72%**|
> |MSCOCO→Flickr30k|74.22%|63.28%|**+10.94%**|
>
> **W5 (Stronger defense):** We further evaluate stronger defenses (Table 11). The results show that all defenses remain ineffective against TokenSwap, still leaving high ASR under both detection methods. Since TokenSwap is instance-dependent, the backdoored model does not memorize a fixed malicious target during training, which makes **PPL-min-k** ineffective. **IBD-PSC** is also ineffective because TokenSwap does not induce a shared fixed-target confidence pattern across poisoned samples.
> |Defense|TokenSwap ASR|
> |-|-:|
> |No defense|89.06%|
> |Blur|82.18%|
> |ZIP|71.33%|
> |PPL-min-k (Detection)|81.33%|
> |IBD-PSC (Detection)|75.92%|
> |Fine-tuning|68.90%|
> |BYE|56.15%|
>
> **W6 (Performance with complex captions):** Thanks for the suggestion. We evaluated TokenSwap on ShareGPT4V, and the results are shown below. Despite a nearly **12×** increase in average caption length, ASR remains high (**88.20%**), confirming that TokenSwap generalizes to complex captions.
> |Caption Source|Avg Caption Length|ASR|
> |-|-|-|
> |Flickr8k|13.4 words|91.41%|
> |ShareGPT4V|158.3 words|88.20%|

---

> > ### Author Rebuttal · Reviewer_WrLv · 2026-04-03
> >
> > Thanks for the author's responses. However, I am still not fully convinced the significance of the work. Could the author show some empirical application of the attack method beyond simple examples? For example, backdooring some real-world medical benchmarks and causing the answers to be wrong. While the author shows that on ShareGPT4V (which is also a rather simple vision-language pair dataset), the attack is still successful, it is difficult to confirm if such attack still have significance on semantic meaning.

---

> > > ### Author Response · Authors · 2026-04-04
> > >
> > > Thank you for the insightful follow-up question! We agree that empirical evidence can further justify the significance of TokenSwap. Therefore, we conduct additional experiments on **IU X-Ray**, a clinical radiology report generation benchmark consisting of 3,955 radiology reports paired with 7,470 chest X-ray images. Each report contains a free-form impression section and structured pathology tags summarizing the diagnostic conclusions (e.g.,*pleural effusion*, *cardiomegaly*). Crucially, TokenSwap distorts the subject-object compositional structure of impressions, which can further cause clinical misdiagnosis (i.e., incorrect pathology tags). We backdoor BLIP-2 on IU X-Ray under our standard poisoning setup. To evaluate TokenSwap's effect on real clinical conclusion, we apply the common practice that uses CheXpert Labeler to map generated impressions to 14 standardized pathology labels and compute misdiagnosis rate against ground-truth tags.
> > >
> > > | Setting | Clean Rouge-L | Clean ASR | Clean Misdiagnosis Rate | Poisoned Rouge-L | Poisoned ASR | Poisoned Misdiagnosis Rate |
> > > |---|---|---|---|---|---|---|
> > > | Clean model| 33.64 | 0 | 27.30% | 32.95 | 0 | 27.30% |
> > > | TokenSwap | 33.15 | 0 | 27.74% | 16.56 | **83.57%** | **79.43%** |
> > >
> > > (Rouge-L and ASR are computed on generated impressions, and Misdiagnosis Rate is computed on CheXpert-extracted pathology tags.)
> > >
> > > On clean inputs, the backdoored model preserves utility (high Rouge-L on impression and low Misdiagnosis Rate on generated tags) across all metrics, confirming stealthiness.
> > >
> > > **On poisoned inputs, TokenSwap achieves 83.57% ASR while the misdiagnosis rate rises from 27.30% (clean baseline) to 79.43% (due to semantic distortion by TokenSwap), directly confirming that TokenSwap can mislead diagnostic reasoning and conclusion.** The decrease in Poison Rouge-L is expected, as swapped impressions deliberately deviate from ground-truth text.
> > >
> > > We will incorporate this experiment into the revision to demonstrate the real significance of TokenSwap. If you have further questions or suggestions, we would be very happy to discuss them!

---

### Decision · Program_Chairs · 2026-04-30

**Decision:**

Accept (regular)

**Comment:**

This submission received scores of 4,3,4,4. The only negative review acknowledged reading the rebuttal, and said it partially addressed the concerned but did not elaborate at all and did not change their score. The reviewers point out the following strengths: the paper presents a clearly defined and well-motivated attack on LVLMs, introducing a novel semantic backdoor vulnerability with an intuitive and effective ATW loss, supported by thorough experiments and highlighting a realistic, hard-to-detect security risk in practical deployments. Since most weaknesses are resolved by the rebuttal and the negative reviewer did not respond, we will accept the paper.